

# A fine-tuned YOLOv5 deep learning approach for real-time house number detection

Murat Taşyürek[1] and Celal Öztürk[2]

[1] Department of Computer Engineering, Kayseri University, Kayseri, Turkey
[2] Department of Computer Engineering, Erciyes University, Kayseri, Turkey

## ABSTRACT

Detection of small objects in natural scene images is a complicated problem due to the blur and depth found in the images. Detecting house numbers from the natural scene images in real-time is a computer vision problem. On the other hand, convolutional neural network (CNN) based deep learning methods have been widely used in object detection in recent years. In this study, firstly, a classical CNN-based approach is used to detect house numbers with locations from natural images in real-time. Faster R-CNN, MobileNet, YOLOv4, YOLOv5 and YOLOv7, among the commonly used CNN models, models were applied. However, satisfactory results could not be obtained due to the small size and variable depth of the door plate objects. A new approach using the fine-tuning technique is proposed to improve the performance of CNN-based deep learning models. Experimental evaluations were made on real data from Kayseri province. Classic Faster R-CNN, MobileNet, YOLOv4, YOLOv5 and YOLOv7 methods yield f1 scores of 0.763, 0.677, 0.880, 0.943 and 0.842, respectively. The proposed fine-tuned Faster R-CNN, MobileNet, YOLOv4, YOLOv5, and YOLOv7 approaches achieved f1 scores of 0.845, 0.775, 0.932, 0.972 and 0.889, respectively. Thanks to the proposed fine-tuned approach, the f1 score of all models has increased. Regarding the run time of the methods, classic Faster R-CNN detects 0.603 seconds, while fine-tuned Faster R-CNN detects 0.633 seconds. Classic MobileNet detects 0.046 seconds, while fine-tuned MobileNet detects 0.048 seconds. Classic YOLOv4 and fine-tuned YOLOv4 detect 0.235 and 0.240 seconds, respectively. Classic YOLOv5 and fine-tuned YOLOv5 detect 0.015 seconds, and classic YOLOv7 and fine-tuned YOLOv7 detect objects in 0.009 seconds. While the YOLOv7 model was the fastest running model with an average running time of 0.009 seconds, the proposed fine-tuned YOLOv5 approach achieved the highest performance with an f1 score of 0.972.

## INTRODUCTION

The quality of geographic information systems (GIS) developed to store, analyze, and display spatial data depends on the accuracy of the data it contains (*Cooperative & Collins, 1988*; *Tasyurek, 2022*). The quality and readability of the image data sets used

Corresponding author
Murat Taşyürek,
murattasyurek@kayseri.edu.tr

in creating an address map are very important (*Ulutaş Karakol, Ataman & Cömert, 2021*). Detecting house numbers from natural scene images containing spatial location information (*Visin et al., 2015*) and processing them with their locations accelerates the address infrastructure (*Öztürkçü & Leyla, 2020*). The natural scene image is the raw form of the momentary image of nature or the environment. The most common source used to obtain house numbers from images is Google Street images, which consist of coordinated panoramic images taken with 360° (*Vandeviver, 2014*). Door numbers from street views detecting and reading (*Asif et al., 2021*) is a computer vision problem (*Zuo et al., 2019*; *Kulikajevas, Maskeliunas & Damaševičius, 2021*) that falls under the category of natural scene text recognition (*Fischler & Firschein, 2014*). Character recognition in images in natural scenes is a complicated problem due to the variability of light, background clutter, severe blur, inconsistent resolution, and many other factors. In addition to these properties, there are deteriorations in the characters and numbers in street view photographs with the effect of natural events.

In recent years, deep learning method has been widely used in image classification, object tracking, pose estimation, text detection and recognition, visual salience detection, action recognition, and scene tagging (*Alzubaidi et al., 2021*; *Bashir et al., 2021*; *Pal & Pradhan, 2023*; *Atasever et al., 2022*). Deep neural networks, deep belief networks, recurrent neural networks and convolutional neural networks are the methods frequently used in deep learning (*Garcia-Garcia et al., 2018*). Among these methods, it has been found that convolutional neural networks (CNN) show high performance in image classification (*Khan et al., 2020*; *Dönmez, 2022*). The CNN model takes its name from the linear mathematical operation between matrices called convolution (*O'Shea & Nash, 2015*; *Maass & Storey, 2021*; *Terzi & Azginoglu, 2021*). The CNN model consists of a multi-layer structure including a convolutional layer, non-linear layer, pool layer and fully connected layer (*Albawi, Mohammed & Al-Zawi, 2017*).

Identifying characters and numbers from natural images is one of the classification problems in computer vision. In the literature, studies on detecting house numbers from street images with CNN models show very high performance in image classification (*Goodfellow et al., 2013*; *Visin et al., 2015*).

In this study, classic CNN models such as Faster R-CNN, MobileNet, YOLOv4, YOLOv5 and YOLOv7 were applied in a CNN-based system designed to detect house numbers from images obtained in real-time with spatial location. However, sufficiently successful results could not be obtained, especially due to the small and variable depths of the house number objects in the image.

Training on more datasets is a solution to improve the performance of CNN-based deep learning models, but collecting large amounts of data imposes a time and financial burden. On the other hand, a fine-tuning method has been widely used in recent years to improve the performance of deep learning models (*Amisse, Jijón-Palma & Centeno, 2021*). Fine-tuning is to increase the model's success by making adjustments on deep learning models (*Subramanian, Shanmugavadivel & Nandhini, 2022*). One of the commonly used fine-tuning methods in the literature is to remove the last layer of the model, the softmax layer, and replace it with its classifier layer. Another fine-tuning method is to change

the value of the parameters, also called hyperparameters, which affect the performance of the models (*Öztürk, Taşyürek & Türkdamar, 2023*). On the other hand, freezing the layers' weights in the previously trained model is a common fine-tuning practice. In this study, a new fine-tuning technique is proposed to improve the performance of deep learning-based models. The proposed technique includes updating the softmax layer, multi-scale training (*Rath, 2022*) and performing the training process with a low learning rate (*Yu, 2016*) rate. The proposed approach's main contributions within this study's scope are presented below.

### Contributions

- A new CNN-based approach is proposed for house number detection with the location in real-time.
- The proposed approach has been tested on real natural scene images taken from Kayseri Metropolitan Municipality.
- In the proposed approach, the performances of Faster R-CNN, MobileNet, YOLOv4, YOLOv5 and YOLOv7 models, which are widely used as CNN models, are examined.
- A fair evaluation was made by comparing Faster R-CNN, MobileNet, YOLOv4, YOLOv5 and YOLOv7 models designed in different structures on a single platform (PyTorch).
- A new fine-tuning technique is proposed to improve the performance of classical CNN-based deep learning models in house number detection.
- The proposed fine-tuned YOLOv5 approach can detect house numbers from natural scene images with a high f1 score of 0.972 in an average of 0.015 s.

### Scope and outline

- Hyperparameter optimization to improve accuracy performance in house door number detection is out of the scope of this study.

The rest of this article is organized as follows: Section 2 presents the related work. Section 3 gives about basic concept with CNN models. Section 4 presents the proposed approach. In Section 5, experimental evaluations are presented. Section 6 presents conclusions and future works.

## RELATED WORKS

CNN method, which is one of the deep learning methods, has been widely used in different fields such as computer networks (*Gu et al., 2018*), image detection (*Chauhan, Ghanshala & Joshi, 2018*) and disease classification (*Lu, Tan & Jiang, 2021*) in recent years. The image classification process with CNN can be done by creating a custom CNN structure or using CNN models with a fixed structure. As an example of custom CNN models, *Wei et al. (2018)* proposed a new technique using the CNN model to effectively and robustly detect multifaceted text in natural scene images. *He et al. (2016)* presented a system for scene text detection by proposing the Text-CNN model, which focuses on extracting text-related regions and features from image components. *Jia et al. (2018)* proposed a CNN-based approach to detect handwritten texts from images of whiteboards and handwritten notes. *Garg et al. (2019)* stated that they detected high performance in MNIST dataset by creating

an efficient CNN model with multiple convolutions, ReLu and Pooling layers. _Athira et al. (2022)_ suggested using a special CNN model for character classification in container identity detection and recognition.

The model developed by _LeCun et al. (1999)_ as LeNet-5 for handwriting and machine-printed character recognition in the 1990s is considered the first successful application of convolutional networks. LeNet-5, a 7-level convolutional network, was developed to recognize handwritten numbers in 32x32 pixel grayscale input images. When it is desired to analyze higher resolution images with the LeNet-5 method, the level of the convolutional network is insufficient (_Paul & Singh, 2015_). AlexNet (_Krizhevsky, Sutskever & Hinton, 2012_) (ImageNet) developed in 2012 produced more successful results than all previous CNN models. CNN models have been continuously developed to achieve higher accuracy and faster results (_Alom et al., 2019_). ZFNet (_Fu et al., 2018_) in 2013, GoogLeNet (_Sam et al., 2019_) and VGGNet (_Simonyan & Zisserman, 2014_) in 2014, ResNet (_Gao et al., 2021_) in 2015 were developed.

The developed CNN models are successful in feature extraction and classification in single-object image analysis but not sufficiently successful in multi-object image analysis. For this reason, _Girshick et al. (2014)_ proposed the R-CNN method to overcome the multi-object problem. The R-CNN divides the image into approximately 2,000 regions and searches within the region with CNN. The computational cost of the R-CNN method is high in terms of time. _Girshick (2015)_ developed the Fast R-CNN method that works faster to eliminate the problem of R-CNN running slow. _Julca-Aguilar & Hirata (2018)_ suggested using the Faster R-CNN algorithm as a general method for detecting symbols in handwritten graphics. _Nagaoka et al. (2017)_ developed a model for text detection based on Faster R-CNN that can be trained in an end-to-end coherent manner. R-CNN algorithms use regions to localize the object within the image. The CNN-based YOLO (You Only Look Once) method, which examines parts of the image likely to contain the object rather than thinning the region, was developed by _Redmon et al. (2016)_. The YOLO method has produced more successful results than many object detection methods used in real-time object tracking. For example, _Li et al. (2018)_ used the YOLO model to detect steel strip surface defects in real-time. _Rahman, Ami & Ullah (2020)_ suggested using the YOLO model for an automatic reverse vehicle detection system from road safety camera images. _Pei & Zhu (2020)_ developed the YOLO model for real-time text detection and recognition. _Taşyürek & Öztürk (2022)_ proposed a two-stage deep learning model using only the YOLOv4 model to detect house numbers from natural scene images. However, in the approach, real-time object detection was not performed, and the location data of the objects on the earth was not captured.

In addition, YOLO models have been constantly being improved. YOLOv5 was developed by _Jocher et al. (2020)_. _Kim et al. (2022)_ examined the object detection and classification performances of YOLOv4 and V5 models on the Maritime Dataset and showed that the YOLOv5 model showed superior object detection performance compared to the YOLOv4 model. On the other hand, _Taşyürek (2023)_ has proposed a new approach called ODRP, which uses map-based transformation and deep learning models to detect

street signs with their real locations on Earth from EXIF format data. In the proposed ODRP approach, the YOLOv5 model outperformed the YOLOv6 model in object detection.

In recent years, the fine-tuning technique has been widely used to increase the classification and segmentation performance of CNN-based deep learning methods (*Pham, 2021*; *Xu et al., 2021*). For example, *Kaya & Gürsoy (2023)* proposed a transfer learning-based deep learning approach with fine-tuning mechanisms to classify COVID-19 from chest X-ray images. They used the MobileNet V2 version as the CNN model, and the proposed model achieved an average accuracy of 97.61% with fine-tuning. *Akshatha et al. (2022)* examined the performance of the Faster R-CNN and SSD models fine-tuned for human detection from air thermal images. After fine-tuning, the mAP metric of the Faster R-CNN model increased by 10%, while the mAP metric of the SSD model increased by 3.5%. *Salman et al. (2022)* proposed the fine-tuned YOLO model for an automated prostate cancer grading and diagnosis system. Thanks to the fine-tuning technique they suggested, the proposed method achieved 97% detection and classification success.

In this study, firstly, classic Faster R-CNN, MobileNet, YOLOv4, YOLOv5 and YOLOv7 models were applied for a CNN-based system that detects house numbers with spatial locations from natural images in real-time. However, satisfactory results could not be obtained due to the small size and variable depth of the house plate object in the raw images. A new approach using the fine-tuning technique is proposed to improve the object detection performance of the CNN-based system.

## BASIC CONCEPTS

Deep learning has become a prevalent subset of machine learning because of its high classification performance across many data types (*Raschka & Mirjalili, 2017*; *Zhang et al., 2017*). One of the most impactful deep learning methods for image classification is the convolutional neural network (CNN) method. CNN is a deep learning algorithm generally used in image processing and takes images as input (*Wang et al., 2017*; *Nasir, Khan & Varlamis, 2021*). This algorithm, which captures and classifies the visual features with different operations, has been widely used in recent years (*Barzekar & Yu, 2022*). CNN-based Faster R-CNN, MobileNet and YOLO models used in this study are presented below.

### R-CNN

R-CNN architecture detects classes of objects in images and their bounding boxes. In the R-CNN model, features that are candidates to be objects in the visual are determined by selective search. In selective search, which works with the hierarchy from small to large, small regions are determined first. Then, two similar regions are merged, and a new larger region emerges. This process continues recursively. In each iteration, more significant regions occur, and the objects in the image are clustered. After about 2,000 regions are determined, each is individually entered into a CNN model, and their classes and bounding boxes are estimated. Specific region candidates for R-CNN are determined by selective search. These district candidates each enter the CNN networks as inputs. At the end of this region nomination process, approximately 2,000 regions emerge, and 2,000 CNN networks

are used for these 2,000 regions. The object class in SVM models and bounding boxes in regression models are determined using the features obtained from CNN networks. The R-CNN model has the following disadvantages:

- Each image needs to classify 2,000 region suggestions. Therefore, it takes a lot of time to train the network.
- It also requires a lot of disk space to store the feature map of the region recommendation.

The backbone of R-CNN models can be changed. AlexNet, VGG 16 or ResNet 50 can be selected as the backbone of the R-CNN. The default backbone of the R-CNN model developed in the PyTorch is ResNet 50 (*Rath, 2021*). The ResNet 50 model consists of 50 layers, including 1 MaxPool layer, one average pool layer and 48 convolutional layers.

R-CNN architecture (*Girshick et al., 2014*) has been developed since it cannot be easily detected with CNN in images with multiple objects. Ross Girshick developed the Fast R-CNN method, which works faster, to eliminate the problem of R-CNN running slow (*Girshick, 2015*). The fast R-CNN model takes all image and region suggestions as input in feed-forward CNN architecture. Also, the Fast R-CNN model combines the ConvNet, Role Pool, and classification layer of the R-CNN model in a single structure. This eliminates the need to store a feature map and saves disk space. It also uses the softmax layer instead of the SVM method in region recommendation classification, which has proven faster and produces better accuracy than the SVM method.

On the other hand, Faster R-CNN were introduced by *Ren et al. (2015)*. In the Fast R-CNN model, the bottleneck is the selective search method for the R-CNN architecture. The region proposal network (RPN) is used instead of the selective search method in the Faster R-CNN model. In this model, the image is first transferred to the backbone network. This backbone network creates a convolutional feature map. This feature map is forwarded to the region recommendation network (RPN). Returns object candidates along with candidate scores objectness using the RPN feature map. Then, The ROI pooling layer resizes the regions to a fixed size. Finally, it feeds the regions to the fully connected layer for classification. Regarding computational cost, Faster R-CNN is faster than R-CNN and Fast R-CNN (*Ren et al., 2015*). In addition, the Faster R-CNN model achieves better mean average precision value than R-CNN and Fast R-CNN models. This study used the Faster R-CNN model, a more successful method than R-CNN and Fast R-CNN methods.

## MobileNet

MobileNet is a CNN-based deep learning model designed for mobile and embedded computer vision applications. The MobileNet (V1) was introduced by *Howard & Zhu (2017)*. MobileNet is a simple and efficient deep learning model (*Michele, Colin & Santika, 2019*). It is widely used in real-time applications due to its low computational cost (*Verma & Srivastava, 2022*; *Edel & Kapustin, 2022*).

The basis of MobileNetV1 is deeply detachable convolutional structures to create lightweight deep neural networks. Deep convolution applies a single filter to each input channel in this release. Point convolution then uses the $1 \times 1$ convolution to combine the outputs of the deep convolution. A standard convolution filters the inputs and combines

them into a new set of outputs in a single step. MobileNet has 28 layers. The model takes an image with dimensions $224 \times 224 \times 3$ as input. On the other hand, the MobileNet model continued to be developed by adding new features. In 2018, the MobileNet V2 was introduced by *Sandler et al. (2018)*. The MobileNet V2 has been developed to overcome the bottlenecks in the intermediate inputs and outputs of the V1 model. Thanks to the improvements made, the Mobilenet V2 model has achieved faster training and better accuracy than the V1 model. On the other hand, the following model version, MobileNet V3, is widely used in the image analysis capabilities of many popular mobile applications.

In this study, the MobileNet V3 version was used because it stands out with its low computation cost in real-time systems.

## YOLO

The YOLO approach takes its name from the words "You Only Look Once", which means you only look once (*Redmon et al., 2016*). The YOLO approach can predict at a glance what the objects in the image are and where they are *Sarkar & Gunturi (2022)*. With the YOLO method, high accuracy can be achieved most of the time, and it also works in real-time, which has been frequently preferred in recent years due to its capabilities (*Du, 2018*). The algorithm "looks only once" at the image in the sense that it only requires one forward propagation pass through the neural network to make the prediction. After non-maximum suppression (which allows the object detection algorithm to detect each object only once), it outputs the recognized objects along with the bounding boxes. With YOLO, a single CNN simultaneously predicts multiple bounding boxes and the class probabilities for those boxes. YOLO can work on full images and directly optimize detection performance.

The YOLO algorithm performs these operations using the CNN model. The architectural structure of the YOLO model consists of 24 convolutional layers, followed by two fully connected layers (*Redmon et al., 2016*). The architecture uses the $7 \times 7$ ($S \times S$) grid structure. It takes $448 \times 448 \times 3$ images as input data. Architecture produces output in size $7 \times 7 \times 30$.

The YOLO approach has been continually developed. YOLO V1 architecture, the first version developed by *Redmon et al. (2016)*, because the output layer is a fully linked layer, the YOLO training model only supports the exact input resolution during testing as the training image. To eliminate the shortcomings of the YOLO V1 version and continue its success, the more accurate, faster, and more powerful YOLO v2 architecture, which can recognize 9,000 objects, was introduced by *Redmon & Farhadi (2017)*. Developed by *Redmon & Farhadi (2018)* in 2018, the YOLOv3 model is more complex than the previous model. The YOLOv3 architecture allows changing the size of the model's structure, allowing the speed and accuracy of the model to be changed. In 2020, the YOLOv4 version was introduced by *Bochkovskiy, Wang & Liao (2020)* as an object recognition method with optimum speed and accuracy. A practical and powerful object detection model is proposed in the YOLOv4 release. YOLOv4 aims to find the best balance between input network resolution, number of convolutional layers, number of parameters, and number of layer outputs (filters).

On the other hand, Jocher developed the YOLOv5 model in 2020 (*Jocher et al., 2020*). Unlike the V4 model, the YOLOv5 model is run in Pytorch. Studies (*Jiang et al., 2022*; *Fang*

*et al., 2021*) have shown that the YOLOv5 model produces more successful estimations and less computational cost than the V4 model. While previous versions of YOLO were written in the C programming language, YOLOv5 was written in the Python programming language. Thus, installing and integrating YOLOv5 into IoT devices has become more accessible. YOLOv5 is only 27 MB, while YOLOv4 using Darknet is 244 MB. Compared to YOLOv4's Darknet community, YOLOv5's Pytorch community is more populated, indicating that more contributions will be made and more significant potential for future growth. It is challenging to accurately compare the performance of the YOLOv4 and YOLOv5 methods, which use two different languages and frameworks. But over time, under the same conditions, the YOLOv5 method has proven itself by showing higher performance than the YOLOv4 method and receiving more support from the computer vision community.

In addition, a new version of the YOLO model, the YOLOv7, was released in 2022 (*Wang, Bochkovskiy & Liao, 2022*). YOLOv7 uses anchor boxes to detect a broader range of object shapes and sizes than previous versions. YOLOv7 also has a higher resolution than previous versions. While other models process images at $416 \times 416$ resolution by default, the YOLOv7 model processes images at $608 \times 608$ by default. Thanks to this default image size, the YOLOv7 model detects smaller objects and gives it higher accuracy overall (*Kundu, 2023*).

In this study, the performances of the YOLOv4, V5 and V7 models were examined.

# PROPOSED CNN BASED DEEP LEARNING APPROACH FOR HOUSE NUMBER DETECTION WITH SPATIAL LOCATION IN REAL-TIME

The quality of geographic information systems developed to store, analyze and display spatial data depends on the accuracy of the data it contains. Since address data has been created using natural scene images in recent years, the legibility of the house number characters in the images is very important (*Taşyürek & Öztürk, 2022*). In addition, detecting house numbers from natural images containing location information and processing them with their locations accelerates the address infrastructure. The instance natural scene image containing the number plate is shown in Fig. 1.

The plate with blue background in Fig. 1 is the door number plate. There is the letter 5A on the plate. Address plates are produced in the same standard color and format. The images of the Kayseri used within the scope of this study, obtained in real-time, also contain the location information of the point where the photo was taken. When the house number in these images is detected, the location of the house number is automatically detected. The location of the point where the photo was taken is accepted as the location of the house number. Determining the door number, the essential component of the address infrastructure, and its correct positioning on the map is essential for vital services such as education, hospital and pharmacy. However, when door numbers are determined from natural images with classical methods, errors occur due to eye strain or typing on the keyboard incorrectly. In this study, a new CNN-based approach is proposed to overcome

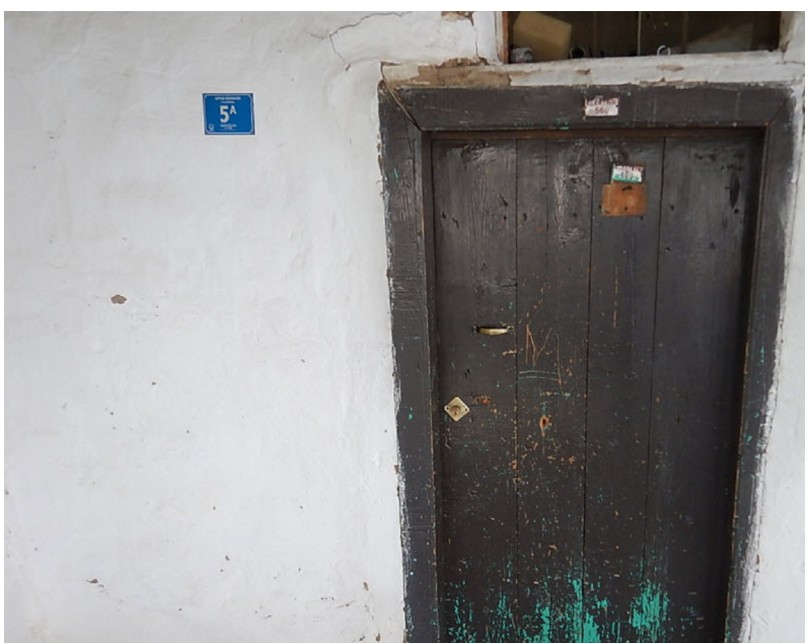

**Figure 1**  Example house number.

these problems and to detect house numbers with their locations in real-time. The flowchart of the proposed system is presented in Fig. 2.

As seen in Fig. 2, firstly, the model must be trained in CNN-based object detection systems. In order to increase the performance of the proposed system, the transfer learning technique was used within the scope of this study. The transfer learning method is frequently used during the training process of CNN-based models (*Zhuang et al., 2020*). Transfer learning model can be expressed as transferring the previously trained and high-performance weights to the new model to be trained (*Weiss, Khoshgoftaar & Wang, 2016*). This way, models that show higher success and learn faster with less training data are obtained using previous knowledge. In the system presented in Fig. 2, the picture containing the house numbers with spatial location is input for door number determination. After the picture is given to the system, the door number in the picture is estimated with the CNN-based deep learning method. Suppose the confidence score of the door number estimated by the deep learning method is above the threshold value. In that case, the system reads the estimated door number, location information in the picture and other attributes and saves this information to the database. '5' was estimated with a confidence score of 0.86, and 'A' was estimated with a confidence score of 0.83 in the sample plate detection presented in Fig. 2. Suppose the confidence score of the door number estimated by the deep learning method is below the threshold value (0.5 was selected for this study). In that case, the system ensures that the user enters the door number, reads the other attribute information from the picture, and saves the data to the database.

Within the scope of this study, firstly, Faster R-CNN, MobileNet, YOLOv4, YOLOv5 and YOLOv7 models, which are widely used as CNN-based deep learning models, were

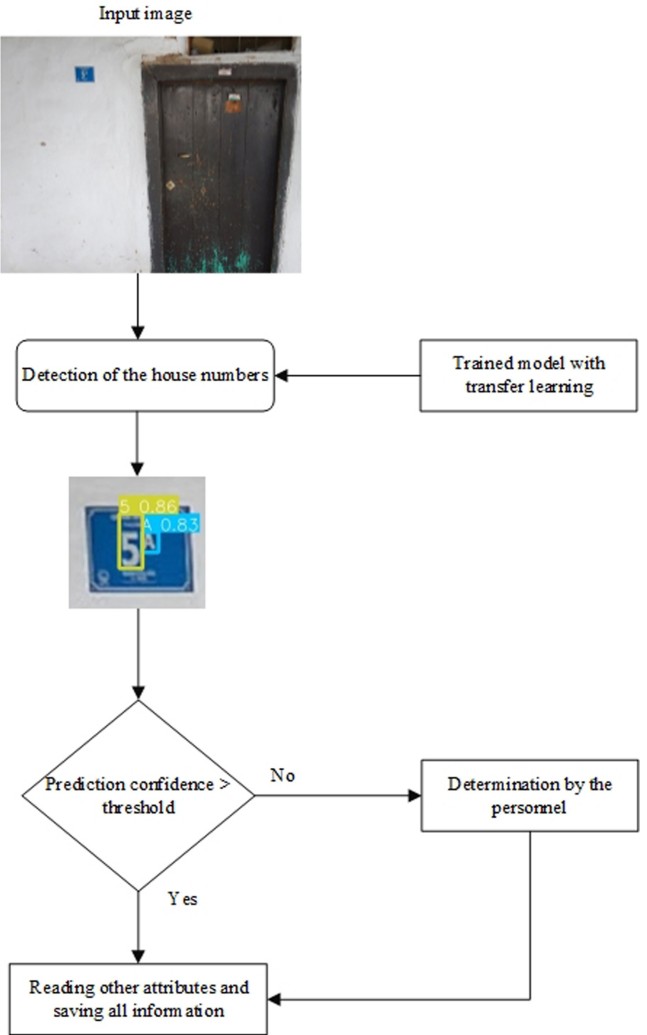

**Figure 2**  **Deep learning-based system architecture.**

applied in the proposed system. The computational costs of YOLO-based models were low, as expected for real-time systems. However, all models could not detect the house numbers and characters sufficiently due to the depth and resolution found in natural images. In order to overcome these problems and improve the object detection performance of CNN-based models, the fine-tuning technique, which has been widely used in recent years, was proposed. Fine-tuning is expressed as increasing the model's success by adjusting deep learning models. There are many fine-tuning types. However, the common and easy-to-use ones can be listed as changing the last layer, reducing the learning rate and multi-resolution training (*Yu, 2016*; *Rath, 2022*). In this study, these three processes were applied. As the first fine-tuning process, the softmax layer of the previously trained network (transferred with the transfer learning technique) was truncated, and a new softmax layer with 14 classes was added instead. As the second fine-tuning process, the learning rate of the models was

reduced, and the models were trained with a learning rate of 0.001. As the final fine-tuning process, the models are trained in multi-resolution. For multi-resolution training, images are automatically resized by +-50% during training with the –multi-scale parameter in YOLOv4, V5 and V7 models. However, this feature is not available on Faster R-CNN and MobilNet models. For Faster R-CNN and MobileNet models, images were resized before fine-tuned training. Results of the classical Faster R-CNN, MobileNet, YOLOv4, YOLOv5 and YOLOv7 models and fine-tuned Faster R-CNN, MobileNet, YOLOv4, YOLOv5 and YOLOv7 models in the proposed approach in following section has been presented.

# EXPERIMENTAL EVALUATIONS

In this section, the experimental performances of Faster R-CNN, MobileNet, YOLOv4, YOLOv5 and YOLOv7 methods are compared for both classical and proposed fine-tuned learning. In the experimental evaluations, the answers to the following questions were examined.

- What are the door number detection performances of approaches using classical CNN models?
- What are the door number detection performances of approaches using fine-tuned CNN models?
- What are the run-times of the approaches?

## Data sets

In this study, natural scene images containing the house numbers with the location were used. 2,664 images were used as training data, and 626 images were used as validation data. To examine the performance of the methods, real images containing 3,627 door numbers and location information in Kayseri province, Sarioglan-Ciftlik district, were used. Detailed information about the images used for testing purposes is presented in Table 1.

The images presented in Table 1 also include locations of door numbers. In other words, while there is a house number on the image, its attributes contain the information at which location the image is taken. The location data in the attribute information is positioned on the map as shown in Fig. 3 using the open-source leaflet library and the open street map base.

The spatial distribution of the dataset is shown in the map image presented in Fig. 3. Since the settlements are more in the town centre, the blue dots showing the location of the house number are more concentrated in the settlements.

## Model settings and performance metrics

The YOLOv5 (*Jocher et al., 2020*) and YOLOv7 (*Wang, Bochkovskiy & Liao, 2022*) models were developed using the PyTorch library. Faster R-CNN (*Rath, 2021*), MobileNet (*Wang, 2019*) and YOLOv4 (*Yiu, 2021*) versions developed with Pytorch architecture were used to compare the methods under equal conditions. All methods were trained by setting the epoch value to 300. Experimental studies have been analyzed using Python 3.9 version on the computer with Intel Core i7-9700 3.0 GHz, 32 GB RAM and 12GB NVIDIA. The loss value produced by deep learning models is used to examine the success of the training

**Table 1  Dataset summary.**

| Dataset ID | Street name | Count of images | Size (Megabyte) |
| --- | --- | --- | --- |
| 1 | AHMET TATAR | 54 | 24.4 |
| 2 | BAĞLAR BAŞI | 270 | 122 |
| 3 | BELEDİYE | 212 | 95.9 |
| 4 | CECELİ | 60 | 27.4 |
| 5 | COŞKUN | 34 | 15.3 |
| 6 | CUMHURİYET MEYDANI | 94 | 42.7 |
| 7 | ESEN | 130 | 59.0 |
| 8 | FATİH | 100 | 45.5 |
| 9 | GEMEREK | 254 | 115 |
| 10 | GÖKTEPE | 278 | 125 |
| 11 | GÜLOVA | 160 | 72.6 |
| 12 | HACILAR | 44 | 16.2 |
| 13 | KANUNİ | 72 | 32.5 |
| 14 | KAYSERİ | 1082 | 488 |
| 15 | KOYUN ABDAL | 58 | 26 |
| 16 | KÖMEVİRAN | 119 | 53.7 |
| 17 | MEHMET AKİF ERSOY | 52 | 23.6 |
| 18 | MEHMET KOÇER | 68 | 30.4 |
| 19 | MESCİT | 48 | 21.6 |
| 20 | NİLÜFER | 30 | 13.6 |
| 21 | NURİ DEĞERLİ | 100 | 45.0 |
| 22 | ORHAN GAZİ | 28 | 12.6 |
| 23 | ORUÇ REİS | 24 | 10.8 |
| 24 | PAPATYA | 44 | 20 |
| 25 | SAĞLIK | 212 | 95.4 |

(*Chung et al., 2020*). The decrease in the Loss value during the training and approaching zero indicates the success of the training. The training graphs of the classical and fine-tuned Faster R-CNN, MobileNet, YOLOv4, YOLOv5 and YOLOv7 model are presented in Figs. 4A and 4B, respectively. In Fig. 4, as the epoch value increases, the decrease in the loss value and gradually approaching zero shows the success of the training process. When the classical CNN models were examined in terms of training times, the training time of the Faster R-CNN model was 77.065 h, the MobileNet training time was 11.241 h, the YOLOv4 training time was 32.133 h, the YOLOv5 training time was 12.133 h and the YOLOv7 training time was 7.891 h. As the fine-tuned CNN models were examined in terms of training times, the training times of Faster R-CNN, MobileNet, YOLOv4, YOLOv5 and YOLOv7 were164.123, 23.413, 63.234, 25.149 and 15.982 h, respectively. The YOLOv5 performed a better training loss value at the same epoch value than the other models for both classical and fine-tuned training, as presented in Fig. 4. The loss value decreases

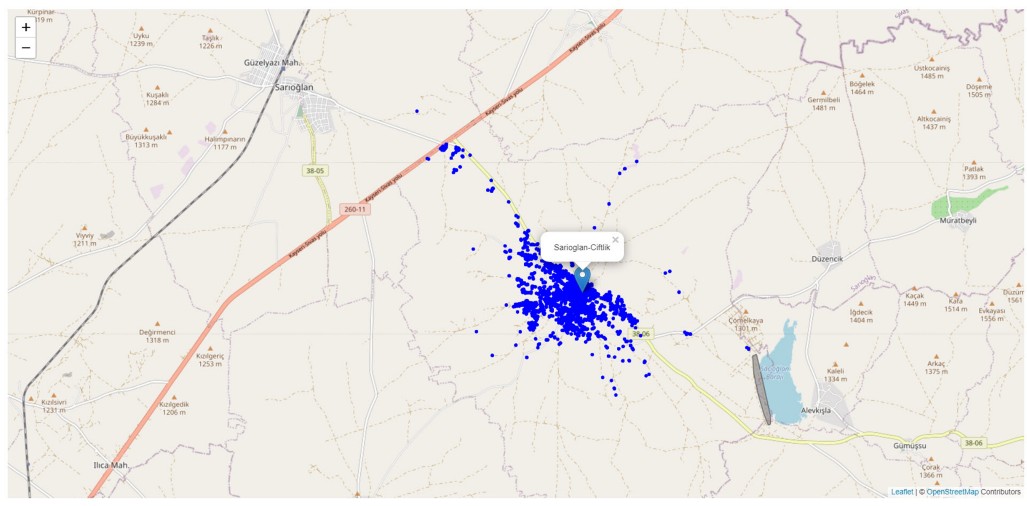

**Figure 3  Spatial locations of the dataset.**

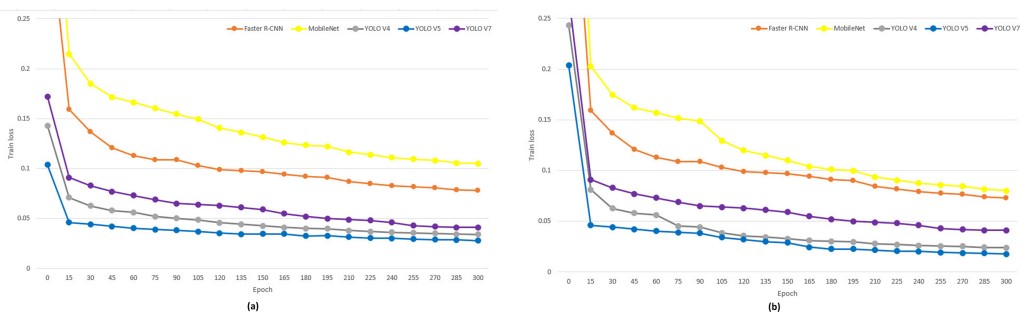

**Figure 4  Train loss of classical and fine-tuned CNN models.**

for a long time since the fine-tuning process reduces the learning rate. In addition, the multi-resolution training increased the training times of the models.

The labelling (annotation) process was done with the LabelImg (*Talin, 2018*) program. The door numbers were analyzed as 14 classes. '0', '1', '2', '3', '4', '5', '6', '7', '8', '9', '/', 'A', 'B' and 'C' were defined and labelled for classes. The YOLOv4, V5 and V7 use .txt files as labelling files, while the Faster R-CNN and MobileNet use .xml files. A single labelling process was made, and the same labels were used for all models by selecting the export formats .txt and .xml.

Performance metrics are used to examine the performance of deep learning models (*Bacchi et al., 2020*; *Teplitzky, McRoberts & Ghanbari, 2020*). These metrics are accuracy, precision, recall and F 1 score. However, in order to calculate these values, true positive (TP), true negative (TN), false positive (FP) and false negative (FN) values must be calculated. If there is an object and detection, this value is accepted as TP. The number of door numbers that the proposed approach detects correctly is the TP value. If there is no object and no detection, this situation is evaluated as TN. If there is a detection by the

model even though there is no object, it is expressed as FP. An object count that cannot be detected by the deep learning model even though it is in the image is referred to as FN.

Accuracy shows how successful the model is in all classes in general and is calculated with Eq. (1).

$$Accuracy = \frac{TP + TN}{TP + FP + TN + FN}.$$

(1)

Precision represents the ratio of the number of correctly classified positive samples to the total number of positive samples and is calculated with Eq. (2).

$$Precision = \frac{TP}{TP + FP}.$$

(2)

Recall measures the model's ability to detect positive samples and is calculated with Eq. (3).

$$Recall = \frac{TP}{TP + FN}.$$

(3)

F1 score is one of the most widely used metrics. F1 score is obtained as a result of using almost all metrics. F1 score is calculated with Eq. (4).

$$F_1 - score = \frac{2 \times Precision \times Recall}{Precision + Recall}.$$

(4)

## Experiments

In this section, experimental comparisons of the approach designed with the CNN-based deep learning model for real-time house number detection are presented. First, the door number detection performances of the classic Faster R-CNN, MobileNet, YOLOv4, YOLOv5 and YOLOv7 models were investigated. Then, the port number detection performances of fine-tuned classic Faster R-CNN, MobileNet, YOLOv4, YOLOv5 and YOLOv7 models were investigated. Finally, the real times of the proposed approaches are presented.

### Door number detection performances of classical CNN models

Within the scope of this experiment, the performance of detecting house numbers in natural scene images of the classic Faster R-CNN, MobileNet, YOLOv4, YOLOv4 and V7 models was compared. Test operations were carried out on 3,627 images. These images contain 20,722 characters (numbers) in total. In order to better examine the performance of CNN models, all benchmark metrics obtained are presented in Table 2.

When the metrics presented in Table 2 are examined, the classical Faster R-CNN, MobileNet, YOLOv4, YOLOv5 and YOLOv7 approaches were able to detect 14,849, 12,532, 17,875, 19,302 and 17,078 as TP, respectively. The TN value was 0 in all models because there was no image without the door number in the dataset. Regarding models' FP values, Faster R-CNN has 3,339, MobileNet has 3,752, YOLOv4 has 2,020, YOLOv5 has 924, and YOLOv7 has 2,780 FP values. On the other hand, Faster R-CNN, MobiletNet, YOLOv4, YOLOv5, and YOLOv7 models have FN values of 5,873, 8,190, 2,847, 1,420 and 3,644, respectively. While TP, FN, FP and FN metrics are used to calculate accuracy,

**Table 2  All metrics of CNN models.**

| CNN Model | TP | TN | FP | FN | Acc. | Pre. | Rec. | F1 S. |
|---|---|---|---|---|---|---|---|---|
| Classic Faster R-CNN | 14849 | 0 | 3339 | 5873 | 0.617 | 0.816 | 0.717 | 0.763 |
| Classic MobileNet | 12532 | 0 | 3752 | 8190 | 0.512 | 0.770 | 0.605 | 0.677 |
| Classic YOLOv4 | 17875 | 0 | 2020 | 2847 | 0.786 | 0.898 | 0.863 | 0.880 |
| Classic YOLOv5 | 19302 | 0 | 924 | 1420 | 0.892 | 0.954 | 0.931 | 0.943 |
| Classic YOLOv7 | 17078 | 0 | 2780 | 3644 | 0.727 | 0.860 | 0.824 | 0.842 |

precision and recall values, precision and recall values are used to obtain the F1 score value. For this reason, it is necessary to look at the F1 score values to examine the performance of the models. The classic YOLOv5 model has the highest f1 score, with 0.943. The MobileNet model, on the other hand, has the lowest f1 score with 0.677. Faster R-CNN, YOLOv4, and YOLOv7 models achieved f1 scores of 0.763, 0.880 and 0.842, respectively. However, although the YOLOv5 model achieves a better f1 score than other models, the f1 score of other models is lower. In order to better analyze the metric values obtained by classical Faster R-CNN, MobileNet, YOLOv4, YOLOv5 and YOLOv7, house number detection of methods on an image was examined and presented in Figs. 5A, 5B, 5C, 5D, and 5E, respectively.

When Fig. 5 is examined, there is the original version of the image in Fig. 5B. This is because the MobileNet model cannot detect any digits and characters in Fig. 5B. Due to such situations, the performance metric of the MobileNet model was lower. As seen in Fig. 5A, the Faster R-CNN model detected the number '6' with a confidence score of 0.73 but failed to detect the character 'A'. The detection result of the YOLOv4 model is presented in Fig. 5C. The YOLOv4 model could not detect the 'A' character but detected the '6' with a confidence score of 0.66. As seen in Fig. 5D, the YOLOv5 model could not detect the 'A' character but detected the '6' with a confidence score of 0.86. The result of detecting the house number of the YOLOv7 model is presented in Fig. 5E. The YOLOv7 model could not detect the 'A' character like other models, but it did detect the '6'. When Fig. 5 is examined, the model that detects '5' with the highest confidence score is YOLOv5. Because it detects with such a high confidence score, the metric values of the YOLOv5 model are higher than the others. However, none of the classical CNN models could detect the 'A' character. In this study, the fine-tuning technique is proposed to detect undetectable characters, such as the 'A' character and to detect door numbers with higher performance rates. The results of the proposed fine-tuning technique are presented in the following experiment.

### Door number detection performances of fine-tuned CNN models

In this experiment, the performance of fine-tuned Faster R-CNN, MobileNet, YOLOv4, YOLOv4 and V7 models to detect house numbers in natural scene images was compared. As shown in the previous section, classical CNN-based models could not detect house numbers in images with variable depths. The fine-tuning technique has been proposed to overcome these problems and to detect door numbers with higher performance rates. The success of the proposed method was examined on 3,627 real images. All benchmark

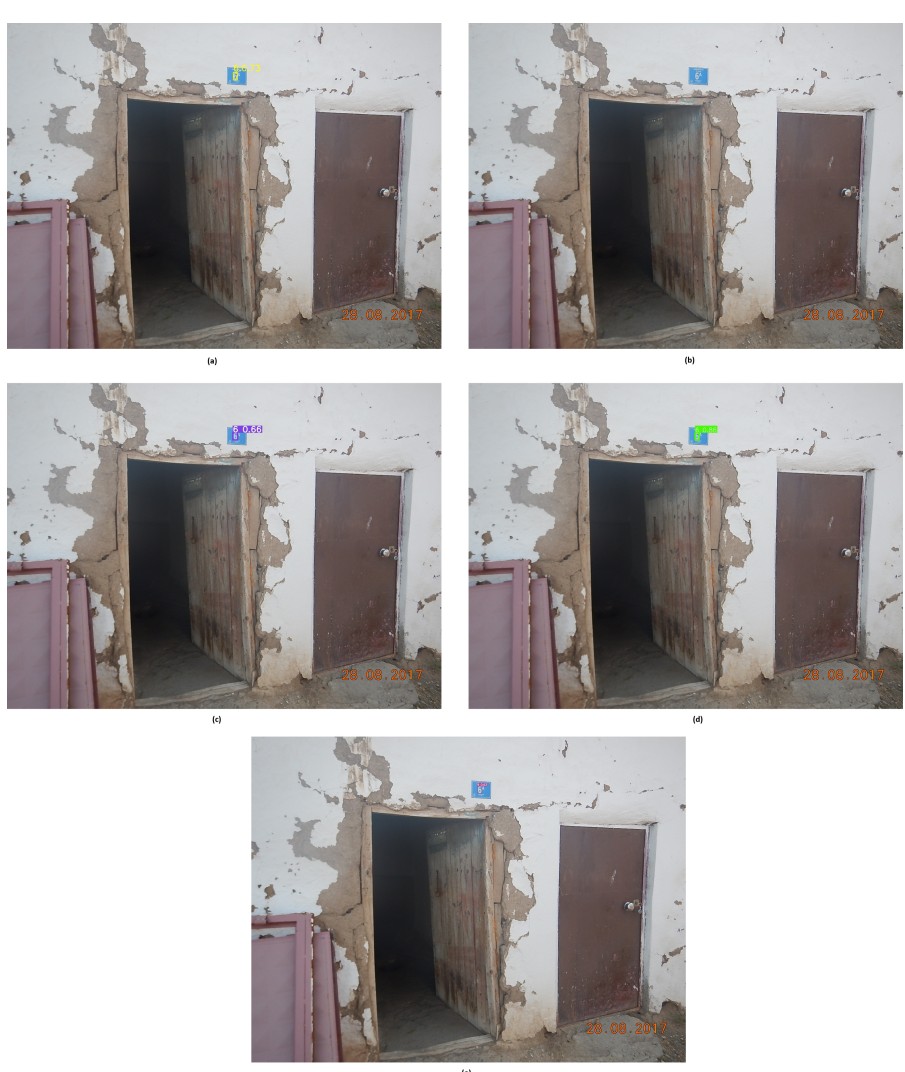

**Figure 5  House number detections by classic CNN models.**

**Table 3  All metrics of CNN models with fine-tuning.**

| CNN Model | TP | TN | FP | FN | Acc. | Pre. | Rec. | F1 S. |
|---|---|---|---|---|---|---|---|---|
| Fine-tuned Faster R-CNN | 17122 | 0 | 2682 | 3600 | 0.732 | 0.865 | 0.826 | 0.845 |
| Fine-tuned MobileNet | 14977 | 0 | 2968 | 5745 | 0.632 | 0.835 | 0.723 | 0.775 |
| Fine-tuned YOLOv4 | 19318 | 0 | 1401 | 1404 | 0.873 | 0.932 | 0.932 | 0.932 |
| Fine-tuned YOLOv5 | 20037 | 0 | 485 | 685 | 0.945 | 0.976 | 0.967 | 0.972 |
| Fine-tuned YOLOv7 | 18345 | 0 | 2224 | 2377 | 0.799 | 0.892 | 0.885 | 0.889 |

metrics showing the performance of the proposed fine-tuned CNN models are presented in Table 3.

When the metrics presented in Table 3 are examined, fine-tuned Faster R-CNN determined 17,122, MobileNet determined 14,977, YOLOv4 determined 19,318, YOLOv5

determined 20,037, and YOLOv7 18,345 determined the door numbers as TP. Fine-tuned Faster R-CNN, MobileNet, YOLOv4, YOLOv5 and YOLOv7 models increased their TP values 2,273, 2,445, 1,443, 735 and 1,267, respectively, compared to their classics. The fine-tuned MobileNet model increased the TP value at the highest rate, with a value of 2,445. Since the classical MobileNet model has a very low TP compared to other models, the highest increase was observed in this model after fine-tuning. The lowest increase in TP values was observed in the fine-tuned YOLOv5 model. This is because the classic YOLOv5 model is also successful. On the other hand, the TN value of all fine-tuned models was 0. When fine-tuned CNN models are analyzed according to FP value, fine-tuned Faster R-CNN, MobileNet, YOLOv4, YOLOv5, and YOLOv7 models have FP values of 2,682, 2,968, 1,401, 485 and 2,224, respectively. If the model finds a number or character, even though there is no number or character, it is considered FP. A low FP value or a decrease in this value compared to the previous model indicates the success of the recommended fine-tuning technique. Thanks to the proposed fine-tuning technique, these models reduced their FP values by 657, 784, 619, 439 and 556, respectively. In addition, these models decreased their FN values amount of 2,273, 2,445, 1,443, 735 and 1,267, respectively, thanks to the proposed method. When the fine-tuned models were examined according to their F1 score values, the order of performance was the same as the classical CNN models. Fine-tuned YOLOv5 has the highest f1 score with 0.972. The fine-tuned MobileNet model, on the other hand, has the lowest f1 score with 0.775. Fine-tuned Faster R-CNN, YOLOv4 and YOLOv7 models achieved f1 scores of 0.845, 0.932 and 0.889, respectively. Thanks to the proposed fine-tuning technique, all CNN models have increased the f1 score performance. In order to better analyze the performances of the proposed fine-tuned CNN models, the method's house number detection on the same image used in classical CNN models was examined. The detection results of the fine-tuned Faster R-CNN, MobileNet, YOLOv4, YOLOv5 and YOLOv7 are presented in Figs. 6A, 6B, 6C, 6D, and 6E, respectively.

As the Fig. 6 is examined, all models except the fine-tuned MobileNet model detected the number '6' and the character 'A' correctly (TP). The fine-tuned MobileNet model caught only 6'. While the classical MobileNet model could not find any object in the same image, the fine-tuned MobileNet model could detect the number '6' thanks to the suggested fine-tuning technique. Fine-tuned Faster R-CNN, YOLOv4, YOLOv5 and YOLOv7 models detected the 'A' character, which they could not detect in their classical state, thanks to the fine-tuning technique. In the input image, the depth of the door plate is high. In other words, the character's size on the door sign is small. Due to variable depth, classical CNN-based models cannot detect the house number successfully enough. In the proposed fine-tuned technique, the models are trained in multi-resolution by changing the size of +-50. Thanks to this multi-resolution training, fine-tuned models can detect more successful house numbers in natural scene images with varying depths than classic CNN models. In addition, as with classical CNN models, the fine-tuned YOLOv5 model is the model that detects house numbers with the highest confidence score. Due to such

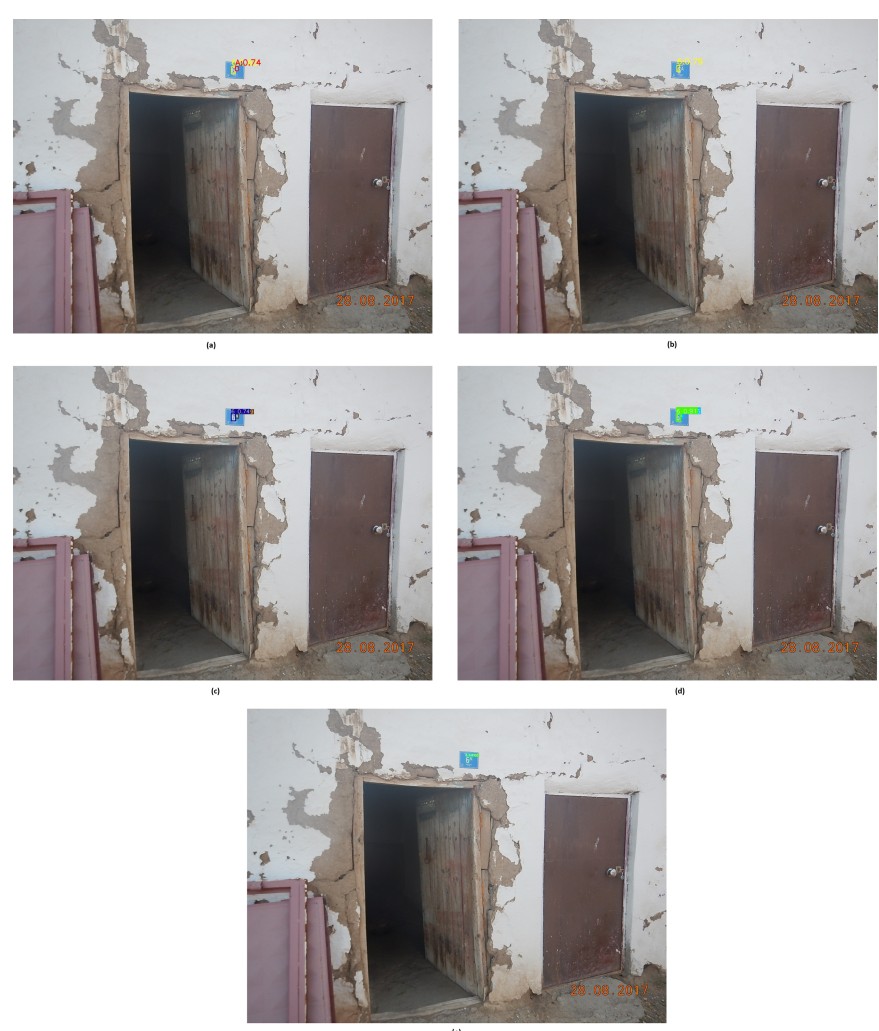

**Figure 6 House number detections by CNN models with fine-tuning.**

successful detections, the performance of the fine-tuned YOLOv5 model is superior to other models.

### Run time of the approaches

In real-time object detection, the computational cost is as important as the estimation performance of the methods. For this reason, the object detection times of the classic Faster R-CNN, MobileNet, YOLOv4, YOLOv5 and YOLOv7 models and the recommended fine-tuned Faster R-CNN, MobileNet, YOLOv4, YOLOv5 and YOLOv7 models were investigated. The PyTorch version of Faster R-CNN (*Rath, 2021*), MobileNet (*Wang, 2019*), YOLOv4 (*Yiu, 2021*), YOLOv5 (*Jocher et al., 2020*) and YOLOv7 (*Wang, Bochkovskiy & Liao, 2022*) models were used to evaluate the models under equal conditions. Models

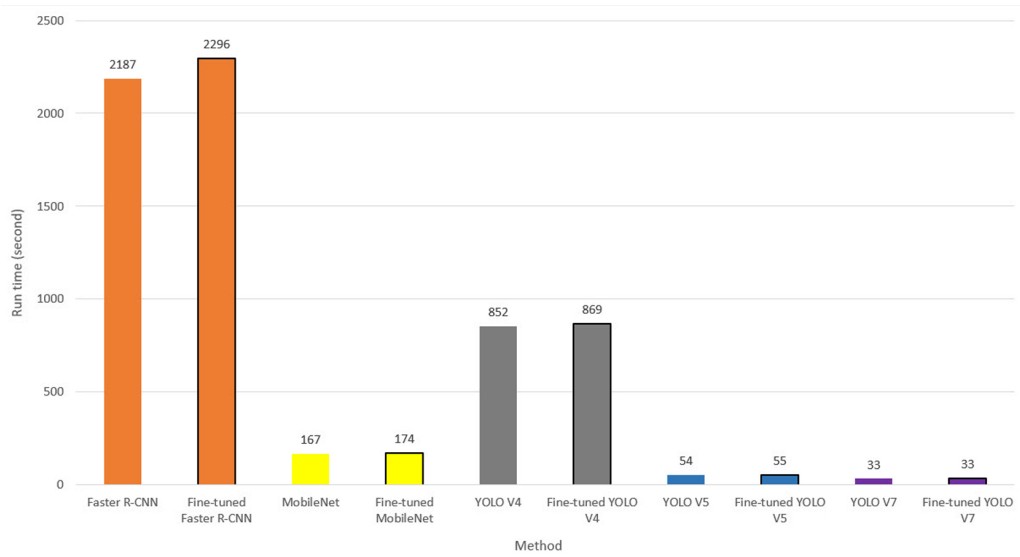

**Figure 7  Run time of CNN models.**

were run for 3,627 images in the dataset. The total running times of models are presented in seconds in Fig. 7.

As seen in Fig. 7, the total run times of the classic Faster R-CNN, MobileNet, YOLOv4, YOLOv5 and YOLOv7 are 2,187, 167, 852, 54 and 33 s, respectively. The total working time of these models with the fine-tuning technique is 2,296, 174, 869, 55 and 33 s, respectively. The Fine-tuned Faster R-CNN model has the highest calculation cost with 2,296 s. Also, the runtime of the classic Faster-RCNN model is higher than the MobileNet and YOLO models. On the other hand, the classic YOLOv7 and fine-tuned YOLOv7 models have the lowest runtime. Classical CNN models detect the house numbers for which an image is found in approximately 0.603, 0.046, 0.240, 0.015 and 0.009 s, respectively. Fine-tuning CNN models detect about 0.633, 0.048, 0.240, 0.015 and 0.009 s, respectively. As a result of the fine-tuning process, the computational cost of the Faster R-CNN model increased by only 0.030 s in object detection. The proposed fine-tuning technique added only 0.020 s to the MobileNet model. This extra computational cost to the YOLOv4 model is 0.005 s. The fine-tuning technique did not affect the average running time of the YOLOv5 and YOLOv7 models. In real-time door number detection, the YOLOv7 method works at least 66 times faster than the Faster R-CNN method, 5 times faster than the MobileNet model, 26 times faster than the YOLOv4, and at least 1.5 times faster than the YOLOv5 model. The YOLOv5 model operates approximately 40 times faster than the Faster R-CNN model, about 3 times faster than the MobileNet model, and about 15 times faster than the YOLOv4 model.

## CONCLUSION

In this study, a CNN-based approach is proposed to detect house numbers with location information from natural images obtained in real-time. The performance of the proposed system has been tested on real images of Kayseri Province. In the proposed method,

classical Faster R-CNN, MobileNet, YOLOv4, YOLOv5 and YOLOv7, which are widely used as CNN models, were used. However, since the depths vary in natural scene images, sufficient successful results could not be obtained. In other words, the distance of the door plate in the image varies. In cases where the door plate is deep, the characters on the plate become challenging to read. The fine-tuning technique has been proposed to achieve higher performance in images with variable depths. The suggested fine-tuned Faster R-CNN, MobileNet, YOLOv4, YOLOv5, and YOLOv7 methods obtained f1 scores of 0.845, 0.775, 0.932, 0.972 and 0.889, respectively. Thanks to the fine-tuning technique of these methods, the f1 score value increased by 0.082, 0.098, 0.052, 0.029 and 0.047, respectively, compared to the classical methods. Among the proposed approaches, the fine-tuned YOLOv5 achieved the highest performance with an f1 score of 0.972. On the other hand, regarding the run time of the proposed fine-tuned based methods, fine-tuned Faster R-CNN, MobileNet, YOLOv4, YOLOv5, and YOLOv7 detect objects about 0.633, 0.048, 0.240, 0.015 and 0.009 s, respectively. The YOLOv7 model is the model that makes the door number the fastest, with an average working time of 0.009 s.

In future studies, it is planned to perform hyperparameter optimization of CNN-based deep learning models with artificial intelligence optimization algorithms.

## ACKNOWLEDGEMENTS

We want to thank Kayseri Metropolitan Municipality for sharing the raw images obtained with their locations in real-time.

### Funding

This study was supported by the Scientific Research Projects Coordination Unit of Kayseri University within the scope of project #FBA-2022-1093. The funders had no role in study design, data collection and analysis, decision to publish, or preparation of the manuscript.

### Grant Disclosures

The following grant information was disclosed by the authors:
The Scientific Research Projects Coordination Unit of Kayseri University within the scope of project: #FBA-2022-1093.

### Competing Interests

The authors declare that there are no competing interests.

### Author Contributions

- Murat Taşyürek conceived and designed the experiments, performed the experiments, analyzed the data, performed the computation work, prepared figures and/or tables, authored or reviewed drafts of the article, and approved the final draft.
- Celal Öztürk conceived and designed the experiments, performed the experiments, analyzed the data, prepared figures and/or tables, authored or reviewed drafts of the article, and approved the final draft.

## Data Deposition

The raw data and code are available in the Supplemental Files.

## Supplemental Information

Supplemental information for this article can be found online at http://dx.doi.org/10.7717/peerj-cs.1453#supplemental-information.

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
