# Peer review of "A fine-tuned YOLOv5 deep learning approach for real-time house number detection"

_PeerJ Computer Science, doi:10.7717/peerj-cs.1453_

## Round 0.1 · original submission · Major Revisions

Authors should carefully review all the comments from the reviewers in particular to focus on the experimental design, and results and discussion sections. Also, it is strongly recommended authors ask a proficient English speaker to proofread the paper before the next submission.

Reviewer 1 ·

Basic reporting

What do you mean by "natural image" here?

English language of the article should be improved - for ex, in the abstract "Faster R-CNN model achieves an average F1 score of 0.762 and the YOLO V4 model an average of 0.879 F1 scores showed that the model obtained an average of 0.945" - which one obtained 0.945 here?. Even it is understandable, not a correct way to express the contents. So check the article for grammatical and sentence fragment errors like this.

In the introduction - "c images taken with 360° Vandeviver" - Vandeviver is a camera or person?

in the introduction - "Variability of light, background clutter, severe blur, inconsistent resolution, etc." what is this? no meaningful sentence.

In the related work section - line number 80, 82, 85, 87, etc. are having reference the author's name two times. Check the article for this kind of error.

Clear language revision is needed.

Experimental design

Technical details like the layer structure of Faster R-CNN and YOLO5 should be included.

Provide the intermediate output of the models for the given figure 1.

Validity of the findings

sample data have been provided; they are robust.

As per the given data validity of the findings matches.

Additional comments

Figure 3 caption is appropriate.

Rename table 1 caption as "Dataset Summary"

What are the class properties of the labels

Reviewer 2 ·

Basic reporting

1. INTRODUCTION part has lag of clarity. It need to be strengthened.
2. Similarly the related work section also need to be improved. There is no much work had been referred.
3. Author says “For this reason, the YOLO model, which is widely used 244 in real-time object detection and achieved successful results in small object detection, has been used as a 245 CNN-based deep learning method in order to increase the prediction performance of the designed system 246 and reduce the computational cost. The results obtained are presented in Chapter .” It is a too lengthy sentence and it has lot of mistakes.
4. Result and discussion part is not up to the level of research. Detailed discussion is needed.
5. Novelty projection in the manuscript is lagging.
6. The overall ambition of the manuscript is low in terms of content.
7. Table 2. All Metrics of CNN Models has lot of unwanted data. It can be minimized.
8. Latest references are needed.
9. What the Figure 3. Locations of Images communicates? Is it required?

Experimental design

The projection of experimental design is very low.

Validity of the findings

no comment

Additional comments

Language needs to be improved.

·

Basic reporting

The manuscript is a well-written and cited related paper in an appropriate manner. The authors well described the data set and empirical evaluation part too.

Experimental design

Designing and evaluation are well-written.

Validity of the findings

It is properly mentioned in the paper. The contributions help the research community.

---

## Round 0.2 · accepted · Accept

Authors have addressed all the comments and it is ready for publication for this journal.

Reviewer 1 ·

Basic reporting

This revision is satisfactory.

Experimental design

Experiments are valid

Validity of the findings

All appropriate.

Reviewer 2 ·

Basic reporting

Good

Experimental design

Modified according to the inputs given

Validity of the findings

Modified according to the inputs given. Check the general comments

Additional comments

Author says "Regarding the run time of the methods, classic Faster R-CNN detects 0.603 sn,
while fine-tuned Faster R-CNN detects 0.633 sn. Classic MobileNet detects 0.046 sn, while fine-tuned
MobileNet detects 0.048 sn. Classic YOLO V4 and fine-tuned YOLO V4 detect 0.235 and 0.240 sn,
respectively". Check the unit of measurement. This is from abstract but in results and discussion author says "On the 507 other hand, regarding the run time of the proposed fine-tuned based methods, fine-tuned Faster R-CNN, 508 MobileNet, YOLO V4, YOLO V5, and YOLO V7 detect objects about 0.633, 0.048, 0.240, 0.015 and 509 0.009 seconds, respectively."